# Validation of Takotsubo Syndrome Scoring System

**DOI:** 10.3390/diagnostics15111314

**Published:** 2025-05-23

**Authors:** Dana Deeb, Ranel Loutati, Louay Taha, Mohammad Karmi, Akiva Brin, Ofir Rabi, Nir Levi, Noam Fink, Pierre Sabouret, Mohammed Manassra, Abed Qadan, Motaz Amro, Benyamin Khalev, Michael Glikson, Elad Asher

**Affiliations:** 1Jesselson Integrated Heart Center, Shaare Zedek Medical Center, Eisenberg R&D Authority and Faculty of Medicine, Hebrew University of Jerusalem, Jerusalem 9103102, Israel; ranellout@gmail.com (R.L.); louayt@szmc.org.il (L.T.); mkarmi@szmc.org.il (M.K.); akivabr@szmc.org.il (A.B.); ofirrab@szmc.org.il (O.R.); levin@szmc.org.il (N.L.); mohammedma@szmc.org.il (M.M.); qadana@szmc.org.il (A.Q.); motaza@szmc.org.il (M.A.); beny31911@gmail.com (B.K.); mglikson@szmc.org.il (M.G.); 2Assuta Medical Centers, Tel Aviv 6329302, Israel; noamfi@assuta.co.il; 3College of French Cardiologists, 13 Rue Niepce, 75014 Paris, France; cardiology.sabouret@gmail.com

**Keywords:** Takotsubo syndrome, Broken-heart syndrome, apical ballooning, scoring system

## Abstract

**Background**: Takotsubo syndrome (TS) mimics acute coronary syndrome in 1% to 3% of patients presenting with chest pain, ECG changes and echocardiographic transient apical wall hypokinesia. **Objectives**: This study aimed to validate a previously developed scoring system on a larger cohort size. **Methods**: Patients admitted to an intensive cardiovascular care unit were divided into three groups: (a) patients diagnosed with TS, (b) females with anterior MI, and (c) other all-comer STEMIs. A 10-point scoring system was used: stressful events (three points), female gender (two points), no history of diabetes mellitus (two points), estimated left ventricular ejection fraction (LVEF) ≤ 40% on admission echocardiography (one point), positive troponin on admission (one point), and no smoking (one point). A *t*-test was applied to the three study groups, sensitivity and specificity testing was performed using the ROC curve method. **Results**: A total of 1150 patients were included in our study: 54 with TS, 97 females with anterior MI and 999 other all-comer STEMIs. Patients in the TS group were predominantly females with a higher rate of stressful events prior to admission, lower rates of diabetes mellitus and smoking, and lower LVEF% systolic function compared to the STEMI cohort. In a multivariate logistic regression analysis, the average TS scoring system was significantly higher in the TS group compared with the anterior STEMI and all-comer STEMI groups (8.3 vs. 5.7 vs. 3.83, *p* < 0.001, respectively) with an AUC of 0.83 for TS score ≥ 8. **Conclusions**: The 10-point TS scoring system is an easy, reliable, and useful diagnostic tool that might help in distinguishing patients with TS and ACS.

## 1. Introduction

Takotsubo syndrome (TS), referred to as “Broken-heart syndrome”, stress-induced cardiomyopathy, or apical ballooning syndrome, is a transient form of cardiomyopathy characterized by reversible left ventricular systolic dysfunction, typically involving apical and mid-segmental hypokinesis or akinesis with preserved basal contractility [1]. It was first described in Japan in 1990. TS has since been recognized worldwide as an important mimic of acute coronary syndrome (ACS), accounting for 1% to 3% of patients presenting with acute coronary syndrome [2] and 0.5% to 0.9% of ST-segment elevation myocardial infarction (STEMI) cases [3]. It affects mainly, but not exclusively, women of post-menopausal age [4]; as suggested by Murakami, T. et al. [5], men mostly present with TS after experiencing physical stress, while women show more correlation with emotionally stressful events.

Despite its transient nature, TS carries significant morbidity and mortality risks, with in-hospital complications such as cardiogenic shock [6], arrhythmias [7], and thromboembolic events occurring in up to 20% of patients [8]. The syndrome predominantly affects postmenopausal women, with emotional or physical stressors acting as common triggers, though the exact mechanisms remain incompletely understood [4].

The pathophysiology of TS is complex and likely multifactorial, involving a cascade of neurohormonal and vascular responses [9]. The prevailing hypothesis centers on excessive catecholamine and stress-related neuropeptide release following acute stress [10], leading to direct myocardial toxicity, microvascular dysfunction, and transient coronary vasospasm. This surge in sympathetic activity [11] may explain why TS often presents with symptoms indistinguishable from ACS, including chest pain, dyspnea, ECG changes (e.g., ST-segment elevations or T-wave inversions), and elevated cardiac biomarkers. Consequently, current guidelines recommend invasive coronary angiography to exclude obstructive coronary artery disease (CAD) in suspected cases, underscoring the diagnostic challenges posed by TS.

However, the reliance on angiography as a diagnostic gold standard presents several limitations. First, it is an invasive procedure associated with non-negligible risks, including vascular complications, contrast-induced nephropathy, and radiation exposure. Second, the overlapping clinical and biochemical profiles of TS and ACS often lead to misdiagnosis, with about 10–100% of TS cases initially labeled as STEMI due to ST elevation on initial ECG [12,13,14], resulting in unnecessary thrombolysis or percutaneous coronary intervention (PCI). Furthermore, even after angiography confirms the absence of obstructive CAD, the diagnosis of TS remains one of exclusion, requiring additional imaging modalities such as cardiac MRI or echocardiography to assess wall motion abnormalities a process that delays definitive diagnosis and prolongs hospital stays. These challenges highlight the urgent need for reliable, non-invasive diagnostic tools that can accurately differentiate TS from ACS early in the clinical course.

To address this diagnostic dilemma, several diagnostic criteria and scoring systems have been developed over the past two decades. The Mayo Clinic criteria [15], widely used in research settings, rely heavily on echocardiographic findings and the exclusion of CAD, limiting their utility in acute decision-making; the InterTAK Diagnostic Score [16], proposed by the International Takotsubo Registry, incorporates clinical variables such as female sex, emotional stress, and the absence of ST-segment depression to estimate the likelihood of TS. While this score demonstrates good specificity, its sensitivity remains suboptimal, particularly in cases triggered by physical stressors or in male patients. Similarly, other proposed scores, such as the NSTE-TS score that requires demographic, clinical, and ECG data to differentiate women with non-ST-elevation Takotsubo syndrome (NSTE-TS) from those with non-ST-elevation myocardial infarction (NSTEMI) [17], emphasizing good diagnostic accuracy in the derivation study cohort, but their generalizability is constrained by small sample sizes and regional variability in patient demographics. Further risk scores have been established to aid in prognosis and risk stratification, such as the PE2RT complication score (assessing pleural effusion, pericardial effusion, right ventricular involvement, and thrombus via CMR), where a higher score (≥2) in TS patients strongly predicted major adverse cardiovascular events (MACEs) during follow-up, offering a simple CMR-based tool for risk stratification in TS [18]. Likewise, the risk score developed by Agrawal et al. [19] using factors like male sex, age >70, comorbidities (hypertension, renal failure), and critical conditions (cardiogenic shock) showed moderate predictive accuracy for mortality. This aligns with the GEIST (German and Italian Stress Cardiomyopathy registry) prognostic score [20], which predicted in-hospital complications in TS patients and may be associated with increased risk of long-term mortality.

Despite these efforts, no single scoring system has achieved universal acceptance, partly due to inconsistent validation across diverse populations and the lack of prospective studies comparing their performance head-to-head.

In 2019, the PLATIS group introduced a novel 10-point scoring system [21] designed to improve the early differentiation of TS from anterior STEMI. Derived from a cohort of 67 TS patients matched with STEMI controls, this score incorporates clinical (e.g., female sex, stress triggers), electrocardiographic (e.g., absence of reciprocal ST changes), and biochemical (e.g., lower troponin-to-creatinine ratio) parameters. Preliminary results suggested high discriminatory accuracy, with an area under the curve (AUC) of 0.92 in the derivation cohort. However, the PLATIS score has yet to be validated in larger, real-world populations, including non-anterior STEMI cases or mixed ACS presentations—a critical gap given that TS can mimic a wide spectrum of ischemic patterns. Moreover, previous TS scores have often been criticized for their limited applicability in emergency settings, where rapid and reliable risk stratification is paramount. A robust validation of the PLATIS score could therefore fill an important clinical void, offering a practical tool to reduce diagnostic delays and unnecessary invasive procedures.

The present study seeks to address these gaps by conducting a comprehensive validation of the PLATIS scoring system in a large cohort divided into three distinct groups: (1) confirmed TS patients, (2) female anterior STEMI patients (a high-risk group often misclassified as TS due to phenotypic overlap), and (3) an all-comer STEMI population (to assess generalizability across ACS subtypes). By comparing the PLATIS score’s performance against existing diagnostic criteria, we aim to determine its real-world accuracy, clinical utility, and potential to streamline the diagnostic pathway for TS. Our findings could pave the way for broader adoption of this tool, ultimately enhancing early recognition, reducing iatrogenic risks, and optimizing resource allocation in acute cardiac care. Furthermore, we acknowledge that urgent coronary angiography remains the gold standard in patients presenting suspected STEMI, and no clinical criteria should delay revascularization. Our score was not designed to replace angiography in high-probability cases of STEMI but to aid in risk stratification in clinically ambiguous scenarios such as borderline troponin or atypical symptoms; we definitely agree that the score utility in clear STEMI equivalents is limited.

This study will provide insights into whether the PLATIS score retains its discriminatory power in subgroups that are traditionally challenging to diagnose, such as male patients or those without overt stressors.

## 2. Methods

A retrospective, single-center observational study was carried out including patients with a diagnosis of TS compared to female patients with anterior STEMI and other all-comer STEMIs.

All female patients that were admitted to the Intensive Cardiac Care Unit (ICCU) at the Share Zedek Medical Center between July 2019 and September 2024 with a diagnosis of TS compared to females with anterior STEMI in addition to all-comer STEMIs regardless of their gender were studied.

Inclusion criteria: age above 18, chest pain suggestive of cardiac origin or requiring admission to rule out STEMI/TS, urgent heart catheterization within 24 h and echocardiography with regional wall motion abnormality/apical hypokinesia (apical ballooning).

Exclusion criteria: chest pain of any reason other than TS/ACS that was diagnosed in the emergency department such as pneumonia, pulmonary embolism, pericarditis, or chest trauma, history of congenital or acquired heart disease including coronary vascular or valvular heart disease determined by history or physical examination, electrocardiography on admission with left bundle branch block, pregnancy, primary cardiomyopathy, history of active malignancy and chemo/hormonal therapy, and current arrhythmia including atrial fibrillation.

### 2.1. Diagnoses

Patients with TS were diagnosed according to the Mayo Clinic diagnosis criteria [15] and met the following criteria: sudden onset of chest pain with ECG changes, reversible balloon-like left ventricular wall motion abnormality at the apex with hypercontraction of the basal segments at left ventriculography or echocardiography, and troponin I changes mimicking acute myocardial infarction. Echocardiography was performed in all patients on day 1 and on days 3–5 of hospitalization. For the purpose of our study, we included only TS patients who had ST-segment elevation on ECG. All patients with the diagnosis of STEMI (STEMI group) met the following criteria according to the European Society of Cardiology/ACCF/AHA/World Heart Federation Task Force for the Universal Definition of Myocardial Infarction [22]: ECG ST elevation in the absence of left ventricular hypertrophy or left bundle branch block, and new ST elevation at the J point in at least two contiguous leads of ≥2 mm (0.2 mV) in men, or ≥1.5 mm (0.15 mV) in women, in leads V2–V3 and/or of ≥1 mm (0.1 mV) in other contiguous chest leads or limb leads. The cardiologist who was in charge at the time of admission determined the diagnosis of STEMI.

### 2.2. TS Scoring System

The TS scoring system was described previously (Asher, E. (2019) [21]); in short, the scoring system is composed of 10 points: stressful events (3 points), female gender (2 points), no history of diabetes mellitus (DM) (2 points), estimated left ventricular ejection fraction (LVEF) ≤ 40% by echocardiography on admission (1 point), positive troponin on admission (1 point), and no smoking (1 point) [21].

### 2.3. Study Groups

Patients were divided into 3 groups: (a) patients diagnosed with TS, (b) females with anterior MI, and (c) other all-comer STEMIs. All were matched for age, cardiovascular risk factors and history of comorbidities. The TS score was calculated for all patients after revision of all the computed files and discharge notes of each of them, taking into consideration their medical and psychological history.

### 2.4. Data Collection

Data included demographics (age, gender, smoking, weight, body mass index); chronic illness (hypertension, diabetes mellitus, dyslipidemia, coronary artery disease, heart failure); previous percutaneous coronary intervention; and previous coronary artery bypass graft. Variables during hospitalization included ECG changes, troponin elevation, percutaneous coronary intervention, coronary artery bypass graft, discharge diagnoses, and discharge recommendations.

All patients were managed according to the discretion of the treating physician. Cardiac troponin was determined upon admission and after 3, 6, 12, 24, and 48 h, and after that every 24 h, and serial 12-lead electrocardiographic findings were recorded. Assessment of LVEF by two-dimensional echocardiography was performed upon admission and during day 3 or 5 after admission. All exams were analyzed by an echocardiography expert (cardiologist). Outcome measures during hospitalization and follow-up were mortality, stroke/transient ischemic attack, development of acute decompensated heart failure, cardiogenic shock, malignant arrhythmias, acute myocardial infarction, urgent revascularization, and TS recurrence.

### 2.5. Ethics

The ethical aspects of the study were addressed and approved by the Shaare Zedek Medical Center Institutional Review Board (IRB). Striving to maintain participants’ anonymity, the IRB approved de-identification during the subsequent database analysis, and informed consent requirements were waived due to the observational nature of the study (approval number SZMC-0212-24).

### 2.6. Statistics

Descriptive statistics were gathered; we performed univariate analyses as appropriate for the categorical and continuous variables to be included in the study and created multivariate models adjusted for age, gender, and previous medical history including cardiovascular risk factors.

Statistical analysis was performed using SPSS Statistics for Windows R 4.3.3 (SPSS Inc., Chicago, IL, USA). Descriptive statistics were used to describe the study population data.

## 3. Results

A total of 1150 patients were included in the study: 54 (5%) in the TS group; 97 (8%) females with anterior STEMI; and 999 (87%) other all-comer STEMI patients.

### 3.1. Patients’ Characteristics

Patients’ characteristics for the TS and anterior STEMI groups are presented in Table 1. The mean age was 69 ± 12.1 years in the TS group, 71.2 ± 12.9 years in the anterior STEMI group, and 72.55 ± 11.36 years in the all-comer STEMI group. Female gender predominated in the TS group (98.1%) and anterior STEMI group (100%), while the all-comer STEMI group was predominantly male (82.9%, *p* < 0.00001). A significantly higher rate of stressful events was documented in the TS group compared to the anterior STEMI and all-comer STEMI groups (68.5% vs. 2.1% vs. 8.3%, respectively, *p* < 0.00001) as seen in Appendix A. Regarding ECG changes, 42.6% of the patients in the TS group had ST-segment elevation on initial ECG. The TS group also showed markedly lower prevalence of DM (20.4% vs. 43.3% vs. 66.5%, *p* < 0.00001) and smoking (11.1% vs. 14.4% vs. 53.7%, *p* < 0.00001) and reduced LVEF < 40% (74.1% vs. 68.0% vs. 35.0%, *p* < 0.00001) compared to the other groups.

### 3.2. Clinical Characteristics

All patients included in our study underwent coronary angiography on admission, showing normal or nonsignificant coronary disease in the TS group. In the anterior STEMI group, all patients underwent PCI to the LAD artery or one of its branches; in the all-comer STEMIs, 40.5% of the patients had intervention to the LAD artery or its branches, and the remaining underwent PCI for the left circumflex artery, right coronary artery or left main artery.

### 3.3. TS Scoring System

The receiver-operating characteristic (ROC) curve for the TS scoring system demonstrated moderate diagnostic accuracy with AUC = 0.78 (95% CI: 0.72–0.84) for differentiating the TS group from the anterior STEMI group when the cutoff was ≥7, with sensitivity of 82% and specificity of 68% (Figure 1), while excellent discriminatory capacity was seen with cutoff ≥ 8; AUC = 0.84 (95% CI: 0.79–0.89) with sensitivity = 78% and specificity = 88% (Figure 1). AUC = 0.83 (95% CI: 0.78–0.88) for differentiating the TS group from the all-comer STEMI group (cutoff ≥ 8, sensitivity = 76%, specificity = 82%), as shown in Figure 2.

The average TS score was significantly higher in the TS group compared to the anterior STEMI and all-comer STEMI groups (8.3 vs. 5.7 vs. 3.83, *p* < 0.001) (Figure 3 and Figure 4) (Table 1).

### 3.4. Outcomes

Mean peak troponin levels were significantly lower in the TS group (5199 ng/L) compared to the anterior STEMI (72,591.41 ng/L) and all-comer STEMI groups (84,363.5 ng/L, *p* < 0.001). Despite higher rates of LV dysfunction, the TS group had no in-hospital mortality (0%), as compared with the anterior STEMI (8.3%) and all-comer STEMI (4.0%) groups (*p* < 0.001). Deaths in the latter groups were attributed to cardiogenic shock, multiorgan failure, mechanical complications of MI, ventricular arrhythmias, or sepsis.

## 4. Discussion

Differentiating Takotsubo syndrome (TS) from acute myocardial infarction (MI), particularly anterior STEMI, remains a significant diagnostic challenge in clinical practice. Both conditions present with acute chest pain, elevated cardiac biomarkers, and ECG abnormalities, often requiring coronary angiography for definitive exclusion of obstructive coronary disease. Some studies have suggested that the prevalence of ST-segment elevation on ECG is less than half in TS patients [13,14], while diffuse T-wave inversions (I, II, V2–V6) and QTc prolongation (>500 ms) are a more characteristic feature of TS [9]. However, early and accurate diagnosis is critical, as management strategies diverge substantially; TS typically requires supportive care, while STEMI demands immediate revascularization.

Beyond ECG findings, recent advances in biomarker profiling may further aid differentiation. Elevated levels of catecholamines (e.g., norepinephrine) and inflammatory markers (e.g., interleukin-6) have been reported in TS compared to STEMI [23,24]. Additionally, microRNA signatures (e.g., miR-1 and miR-133a) show promise as discriminators, with TS patients exhibiting distinct expression patterns [25]. These molecular markers could complement the TS score in future diagnostic algorithms.

The TS score emerges as a valuable tool to address this diagnostic dilemma, offering a quantitative, evidence-based approach for risk stratification. Our analysis demonstrates that a cutoff ≥ 8 provides the optimal balance between sensitivity (78%) and specificity (88%), with a high positive likelihood ratio, making it particularly useful as a confirmatory rule-in tool for stable patients when clinical suspicion is moderate to high. Notably, the score’s performance aligns with the InterTAK Diagnostic Score [7], which similarly integrates clinical features (female sex, emotional stress) but lacks LVEF and smoking components. Our simplified 10-point system may offer better practicality in emergency settings. Furthermore, the score’s specificity for TS versus non-anterior STEMI (AUC = 0.83) suggests utility beyond anterior wall comparisons, though further validation is needed.

Our score was not designed to replace angiography in high-probability cases of STEMI but to aid in risk stratification in clinically ambiguous scenarios such as borderline troponin or atypical symptoms; we definitely agree that the score utility in clear STEMI equivalents is limited and angiography should never be deferred.

While TS demonstrates lower in-hospital mortality (0% vs. 4–8% in STEMI) despite low initial LV dysfunction, emerging data reveal significant long-term risks, including cardiovascular mortality and increased recurrence rates, particularly in patients with psychiatric or neurologic comorbidities [26,27]. A recent study suggested a prognostic value of C-reactive protein (CRP) alongside the InterTAK score [28] in mortality risk stratification for Takotsubo syndrome; an additional study suggested that higher CRP levels at discharge correlate with increased mortality risk, possibly due to inflammation-driven myocardial dysfunction and residual inflammatory risk [29].

Regarding LV function recovery, studies suggest that those with older age, low LVEF or cardiogenic shock at admission have late LV function recovery and reduced short- and long-term survival [29]. Therefore, TS requires long-term multidisciplinary care and long-term follow-up after hospital discharge.

For borderline scores (<7), adjunctive assessments (e.g., echocardiography for apical ballooning or biomarker kinetics) are recommended. Advanced imaging techniques like cardiac MRI with T1/T2 mapping may further clarify ambiguous cases by detecting diffuse edema without late gadolinium enhancement [30]. The limited availability of cardiac MRI presents a practical challenge in real-world clinical settings. This stratified use balances early detection with diagnostic precision, addressing the hallmark challenge of clinically indistinguishable presentations while optimizing resource utilization. While the score’s integration into ACS algorithms can standardize decision-making, clinical judgment remains critical for atypical cases.

Future studies should focus on expanding the cohort to include additional centers with MRI capabilities, prospective validation of the TS score in real-world cohorts, including its integration with AI-assisted ECG analysis [31] or advanced imaging [32], and further exploration of its prognostic value—such as predicting recovery of LV function or risk of recurrence—to strengthen its role in long-term management.

### Limitations

Our study is a single-center retrospective study that lacks long-term follow-up for our patients and might be limited due to missing data. We appreciate that intentionally restricting our TS cohort to only those cases with a definitive diagnosis, confirmed by cardiac MRI, inherently limited the total number of eligible patients. However, we prioritized diagnostic accuracy over quantity to minimize misclassification bias. We definitely agree that expanding the cohort to include additional centers with MRI capabilities would strengthen future studies. The score’s positive predictive value is highly prevalence-dependent; in low-prevalence settings, even scores ≥ 8 may yield false positives, necessitating clinician judgment. Atypical TS variants (e.g., midventricular or basal ballooning) may not be fully captured by the current score, highlighting the need for multimodal assessment. Lastly, if there is any doubt, emergency coronary angiography is still warranted to rule out acute MI and obstruction of the coronary artery.

## 5. Conclusions

In the clinically challenging distinction between TS and MI, our 10-point diagnostic scoring system provides a simple, rapid, and reliable diagnostic tool to augment diagnostic accuracy. While not a standalone solution, its strategic application prioritizing sensitivity (≥7) for screening and specificity (≥8) for confirmation demonstrates high utility in distinguishing TS from ACS, even in women. The system holds promise for implementation in emergency clinical settings, with future potential for automation within electronic health record systems and integration with artificial intelligence-based diagnostic support.

## Figures and Tables

**Figure 1 diagnostics-15-01314-f001:**
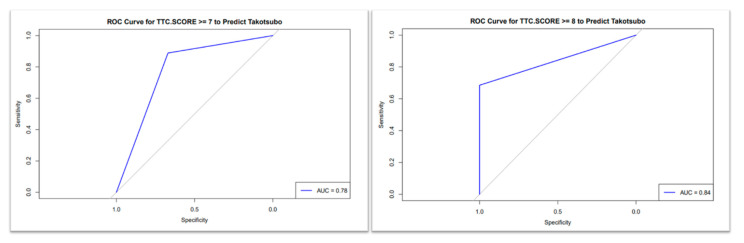
The receiver-operating characteristic (ROC) curve in differentiating Takotsubo syndrome from anterior STEMI; a TS score cutoff ≥ 7 (**right**) demonstrated moderate diagnostic accuracy, with an area under the curve (AUC) of 0.78, while cutoff ≥ 8 (**left**) demonstrated excellent diagnostic performance, with an area under the curve (AUC) of 0.84.

**Figure 2 diagnostics-15-01314-f002:**
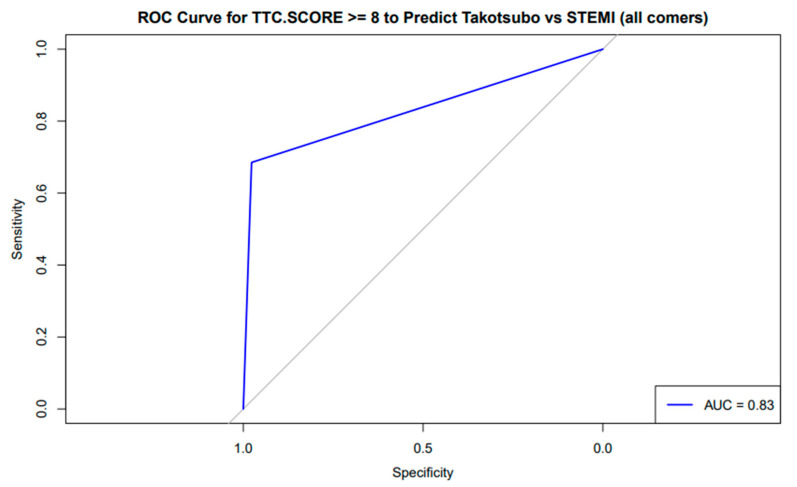
The receiver-operating characteristic (ROC) curve evaluating the TS score (cutoff ≥ 8) for discriminating Takotsubo syndrome (TS group) from all-comer STEMI demonstrated an area under the curve (AUC) of 0.83.

**Figure 3 diagnostics-15-01314-f003:**
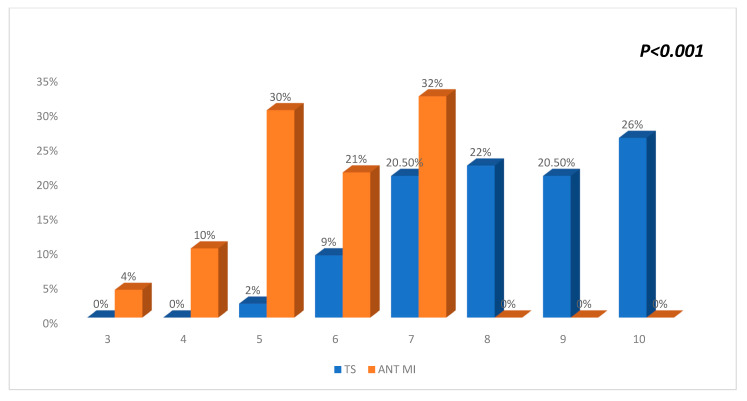
Comparison of the TS score between the TS group (blue) and the anterior STEMI group (orange), showing significantly higher scores with the TS group, suggesting the diagnosis of TS.

**Figure 4 diagnostics-15-01314-f004:**
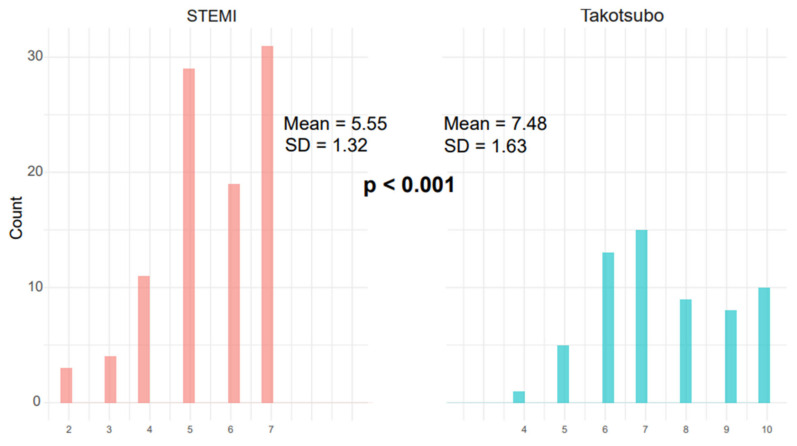
This figure compares the distribution of TS scores between patients with ST-elevation myocardial infarction (STEMI) (red) and Takotsubo syndrome (TS) (blue). The TS group demonstrated significantly higher mean scores (7.48 ± 1.63) compared to the STEMI group (5.55 ± 1.32; *p* < 0.001), with minimal overlap in score distributions.

**Table 1 diagnostics-15-01314-t001:** Patient baseline characteristics.

	Female Anterior STEMI (*N* = 97)	Takotsubo Syndrome (*N* = 54)	All-Comer STEMIs (*N* = 999)	Overall (*N* = 1150)	*p*-Value
**Age**					
Median (Q1–Q3)	71.0 (64.0–83.0)	68.0 (61.3–78.8)	72.0	70.0 (38–106)	0.041
Mean (SD)	71.2 (12.9)	69.0 (12.1)	72.55 (11.36)	70.97 (12.21)	
**Gender**					
F	97 (100%)	53 (98.1%)	171 (17.1%)	321 (27.91%)	0.001
M	0 (0%)	1 (1.9%)	828 (82.9%)	829 (72.0%)	
**Height**					
Median (Q1–Q3)	1.60 (1.57–1.64)	1.60 (1.55–1.65)	1.7 (1.37–1.9)	1.69	0.001
Mean (SD)	1.61 (0.0595)	1.60 (0.0685)	1.7 (0.09)	1.68 (0.011)	
Missing	2 (2.1%)	0 (0%)	4 (0.4%)	6 (0.52%)	
**Weight**					
Median (Q1–Q3)	68.0 (60.0–77.5)	70.0 (56.3–80.0)	80 (45–160)	79.0	0.001
Mean (SD)	70.0 (13.6)	67.9 (15.7)	80.08 (18.02)	78.3 (17.6)	
Missing	2 (2.1%)	0 (0%)	4 (0.4%)	6 (0.52%)	
**BMI**					
Median (Q1–Q3)	26.4 (24.1–29.8)	26.7 (22.2–31.1)	26.67 (14.86–51.65)	26.7	0.012
Mean (SD)	27.1 (4.54)	26.4 (6.09)	27.63 (5.31)	27.4 (5.3)	
Missing	2 (2.1%)	0 (0%)	4 (0.4%)	6 (0.5%)	
**Ethnicity**					
1	83 (85.6%)	43 (79.6%)	888 (88.9%)	1014 (88.2%)	0.001
2	12 (12.4%)	10 (18.5%)	105 (10.51%)	127 (11.0%)	
3	2 (2.1%)	1 (1.9%)	6 (0.6%)	9 (0.78%)	
**Smoking**					
no	83 (85.6%)	48 (88.9%)	463 (46.3%)	594 (51.65%)	0.001
yes	14 (14.4%)	6 (11.1%)	536 (53.7%)	556 (48.34%)	
**Stressful event**					
no	95 (97.9%)	17 (31.5%)	916 (91.7%)	1028 (89.39%)	<0.001
yes	2 (2.1%)	37 (68.5%)	83 (8.3%)	122 (10.61%)	
**DM**					
no	55 (56.7%)	43 (79.6%)	335 (33.5%)	433 (37.65%)	0.00001
yes	42 (43.3%)	11 (20.4%)	664 (66.5%)	717 (62.35%)	
**LVEF < 40%**					
no	31 (32.0%)	14 (25.9%)	648 (64.9%)	693 (60.26%)	0.001
yes	66 (68.0%)	40 (74.1%)	350 (35.0%)	456 (39.65%)	
**Positive troponin on admission**					
no	3 (3.1%)	0 (0%)	66 (6.6%)	69 (6.0%)	0.023
yes	94 (96.9%)	54 (100%)	933 (93.4%)	1081 (94.0%)	
**TS SCORE**					
Median (Q1–Q3)	6.00 (5.00–7.00)	8.00 (7.00–9.75)	3.83	4.00	<0.001
Mean (SD)	5.7 (1.16)	8.3 (1.39)	3.83 (1.53)	3.92 (1.73)	

STEMI = ST-elevation MI; BMI = body mass index; DM = diabetes mellitus; LVEF = left ventricle ejection fraction; TS = Takotsubo syndrome.

## Data Availability

The original contributions presented in this study are included in the article/Appendix A. Further inquiries can be directed to the corresponding author(s).

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
