# Peer review of "Validation of Takotsubo Syndrome Scoring System"

_diagnostics, 2025, doi:10.3390/diagnostics15111314_

Round 1

Reviewer 1 Report

Comments and Suggestions for Authors

Thank you for asking me to review this manuscript.

  1. Suggest using the term "Takotsubo syndrome" instead of "cardiomyopathy" because technically Takotsubo is not a cardiomyopathy.
  2. In my opinion, the main concern is about the sample size. AUC is very sensitive to sample size and event numbers. To determine the number of events necessary to validate a predictive model, at least 100 events and 100 non-events are recommended (Collins GS, Ogundimu EO, Altman DG. Sample size considerations for the external validation of a multivariable prognostic model: a resampling study. Stat Med. 2016; 35: 214-226.). The authors report a lower number of events (54 patients with TS, less than half of what is needed) and non-events (97 patients with anterior STEMI).
  3. It is unlikely that any criteria will allow reliable discrimination of TS with ST-segment elevation on ECG from STEMI at admission. How many TS patients actually had ST-segment elevation on ECG in this study? This was not described clearly. Given the clinical
    impact of delaying revascularisation in STEMI, this study's data does not support the ability of any clinical score to precisely differentiate STEMI and TS to preclude the need for urgent coronary angiography if that is available.
  4. Not sure why all comers of STEMI were included in this study as there were a lot of men in the all comers of STEMI where TS predominantly affects women. This could explain why the score will be low in the all comers of STEMI.

Author Response

Comment 1: Suggest using the term "Takotsubo syndrome" instead of "cardiomyopathy" because technically Takotsubo is not a cardiomyopathy.

Thank you for pointing this out. we  agree with your comment. The term cardiomyopathy was replaced with syndrome in the manuscript.  

Comment 2: In my opinion, the main concern is about the sample size. AUC is very sensitive to sample size and event numbers. To determine the number of events necessary to validate a predictive model, at least 100 events and 100 non-events are recommended (Collins GS, Ogundimu EO, Altman DG. Sample size considerations for the external validation of a multivariable prognostic model: a resampling study. Stat Med. 2016; 35: 214-226.). The authors report a lower number of events (54 patients with TS, less than half of what is needed) and non-events (97 patients with anterior STEMI).

We agree with your comment,  thanks for referring the relevant study. However, In our research we included in the TTC group only cases with confirmed diagnosis of Takotsubo according to cardiac MRI that the patients' underwent following their hospital discharge. Consequently, this stringent inclusion criterion limited the total number of eligible patients.

Comment 3: It is unlikely that any criteria will allow reliable discrimination of TS with ST-segment elevation on ECG from STEMI at admission. How many TS patients actually had ST-segment elevation on ECG in this study? This was not described clearly. Given the clinical impact of delaying revascularisation in STEMI, this study's data does not support the ability of any clinical score to precisely differentiate STEMI and TS to preclude the need for urgent coronary angiography if that is available.

We acknowledge your valid concern regarding the challenge of reliably differentiating TTC from STEMI in patients presenting with ST-segment elevation. As emphasized, urgent coronary angiography remains the gold standard in such cases, and no clinical criteria should delay revascularization when STEMI is suspected.

To clarify our study’s scope, among the takotsubo cohort 42.6% presented with ST segment elevation on initial ECG (now reported in the results section line 536-537). Our score was not designed to replace angiography in high probability cases of STEMI but to aid in risk startification in clinically ambiguous scenarios such as borderline troponin or atypical symptoms, we definitely agree that the score utility in clear STEMI equivalents is limited. In the section of discussion (line 632-635) we clearly explained that in cases of clear ST elevation angiography should never be deferred

Comment 4: Not sure why all comers of STEMI were included in this study as there were a lot of men in the all comers of STEMI where TS predominantly affects women. This could explain why the score will be low in the all comers of STEMI.

We agree with your observation that Takotsubo syndrome (TS) exhibits a well-documented female predominance. However, given that TS is not exclusively limited to women—with male patients representing a clinically significant minority—we intentionally included both genders in our cohort to ensure the generalizability of our findings. According to litereture review, we found that men could present with TS mostly after a physical stress (as mentioned in Murakami T et al 2022) while women showed more correlation with emotional stressful events, according to our score that included physical and stressful events we decided to include men as well to overcome a critical selection bias. (see lines 61-63) 

Reviewer 2 Report

Comments and Suggestions for Authors

The paper " Validation of Takotsubo Cardiomyopathy Scoring System" by Deeb et al. is a good paper. This study attempts to identfy markers to distinguish Takotsubo Cardiomyopath< from anterior wall infarction. 

Takotsubo cardiomypathy is defined as a stress-induced highly symptomatic chest pain event with apical ballooning of the left ventricular apex, but without obstruction of the cornary artery. The pathopysiology is not really clear, but TTC is associated with high endogenous catecholamines. It is not clearly associated with the classical cardiac risk factors but the absence of these risk factors is not a point for the diagnosis for TTC. The left ventricular systolic dysfunktion of the apical and midventricular myocardium gives in the absence of coronary obstruction gives the diagnosis. Above that, the pattern of left ventricular dysfunction does not correspond to the cover the coronary blood supply. This means, that TTC is also not a picture of a myocardial infaction with spontaneous recanalisation of an obstruced coronary vessel with a thrombus of plaque. TTC occurs just completelely independet of coronay vessel supply. TTC patients usually do not hqave the classical coronary risk factos, but the absence of coronary risk factors does not make a point. I personally do not think, that a scoring system for TTC is necessary. It is very clear at the moment of the coronary angiogram, ideally in combination with an LV-Angiogram, after Coronary vessel obstruction is excluded. However, eith ST-Elevations in the ECG you definitively need a coronary aniography. You never would risk to overlook an anterior myocadial infarction because you have used a TTC-Score before. You would not perform an MRT before angiography and you would do an echo only if it is easily possible on the way to the cathlab.

However, this group investigated what distinguishes an anterior myocardial wall infarction from TTC. This is a good way to do that. And may help to investigate the pathophysiology of TTC. 

Therefor, I would suggest to publish this study, although I personally think, that a TTC scoring system is not needed.

However, in the last consequence the authors could more emphasize in their conclusion, that getting the TTC-Score should not delay the immediate coronary angiogram (to not overlook or delay the diagnosis a potential anterior wall infarction)

Author Response

We acknowledge your valid concern regarding the challenge of reliably differentiating TTC from STEMI in patients presenting with ST-segment elevation. As emphasized, urgent coronary angiography remains the gold standard in such cases, and no clinical criteria should delay revascularization when STEMI is suspected. Our score was not designed to replace angiography in high probability cases of STEMI  but to aid in risk startification in clinically ambiguous scenarios such as borderline troponin or atypical symptoms, we definitely agree that the score utility in clear STEMI equivalents is limited (lines 147-152 and lines 683-685). In the section of discussion we clearly explained that in cases of clear ST elevation angiography should never be deferred. To take in cocern that MRI is thought to be the gold standard for confirmation of TTC however its limited availability presents a practical challenge in real-world clinical settings (lines 670-671) 

Reviewer 3 Report

Comments and Suggestions for Authors

In their work, Deeb D. et al. aimed to validate the existing scoring system for Takotsubo cardiomyopathy, comparing values of this scoring system between patients with Takotsubo cardiomyopathy, females with anterior STEMI and all-comers STEMI population.

Major remark is in the Objective of the manuscript. Authors stated: “To validate a previously developed scoring system on a larger cohort size”. Validation would imply having a group of patients with diverse diagnoses, calculate the score that will point to the diagnosis of Takotsubo syndrome or rule out this conditions. The authors performed the reverse – they included patients with known diagnosis and then used the scoring system, which is not the validation. Therefore, validation should provide the diagnostic performance of such a scoring system.

Another major remark is related to this result: “In multivariate logistic regression analysis, the average TTC score was significantly higher in the TTC group compared to the anterior STEMI and all-comers STEMI groups 242 (8.3 vs. 5.7 vs. 3.83, p < 0.001) (Figure 3 and Figure 4)”. In logistic regression, the outcome is binary (yes/no). These results do not relate to binary logistic regression, nor to multivariate logistic regression. Please consult a statistician.

Minor remarks:

Abstract:

Line 24 – Instead of “multivariant” it is correct to say “multivariate”

Line 18-19: It is unclear for what indication the scoring system will be used (in what group of patients).

Line 29-31: TTC patients had “higher rate of stressful event prior to admission and lower rates of diabetes mellitus, smoking and lower LVEF% systolic function” compared to which group(s)?

Comments on the Quality of English Language

Needs significant improvement.

Author Response

We sincerely appreciate your insightful critique regarding the terminology used in our study. To clarify, our primary goal was to evaluate the performance of the existing scoring system by applying It to a larger cohort of confirmed TTC patients and comparator STEMI groups rather than to validate the score in an undifferentiated population. This approach allowed us to rigorously assess the score’s discriminatory power in distinguishing TTS from STEMI after an established definitive diagnosis via CMR which is a critical step before validation in undiagnosed patients.We intended to test the diagnostic consistency within known populations before proceeding to broader validation. We acknowledge your note, that full validation would require application in an undifferentiated cohort with suspected ACS, an eplicited recommendation as a future study.

Minor remarks:

Abstract:

Line 24 – Instead of “multivariant” it is correct to say “multivariate”- corrected 

Line 18-19: It is unclear for what indication the scoring system will be used (in what group of patients)- In our study we suggested using TTC score to aid in risk startification in clinically ambiguous scenarios such as borderline troponin or atypical symptoms (section: introduction lines 147-152)

Line 29-31: TTC patients had “higher rate of stressful event prior to admission and lower rates of diabetes mellitus, smoking and lower LVEF% systolic function” compared to which group(s)? compared to the STEMI cohort (females with anterior STEMI and STEMI all comers group (added to the text) 

Reviewer 4 Report

Comments and Suggestions for Authors

This study investigates a 10 point diagnostic scoring system designed to differentiate Takotsubo cardiomyopathy (TTC) from acute coronary syndrome (ACS). The 10-point scoring system is a rapid, simple, and reliable diagnostic tool, with scores of 8 or higher strongly suggestive of TTC. It demonstrates high utility in distinguishing TTC from ACS even in women. The system holds promise for implementation in emergency clinical settings, with future potential for automation within electronic health record systems and integration with artificial intelligence–based diagnostic support.

I have some minor comments. 

  1. Small number of TTC cases (n=54)
     Takotsubo cardiomyopathy patients comprised only about 5% of the total cohort. This relatively small sample size raises concerns about the statistical power and robustness of the score’s validation, especially for subgroup analysis and external application.

  2. Limited applicability to males and atypical TTC variants
     Since the scoring system includes "female sex" as a scoring item, there is an inherent gender bias. Its diagnostic accuracy in male patients or atypical TTC variants (e.g., midventricular or basal ballooning types) remains unclear and unvalidated.

  3. Lack of head-to-head comparison with existing scores
     The study does not adequately compare the proposed scoring system with other established diagnostic tools such as the InterTAK Diagnostic Score, making it difficult to determine its relative clinical superiority. Could authers calculate InterTAK score in patients of this study?

Author Response

We sincerely appreciate your insightful observation and crucial remarks. 

Comment 1: Small number of TTC cases (n=54). Takotsubo cardiomyopathy patients comprised only about 5% of the total cohort. This relatively small sample size raises concerns about the statistical power and robustness of the score’s validation, especially for subgroup analysis and external application.

Regarding the sample size limitations in our study, to clarify, we intentionally restricted our TTC cohort to only those cases with a definitive diagnosis, confirmed by cardiac MRI. This rigorous approach was necessary to ensure diagnostic certainty, though it inherently limited the total number of eligible patients. This stringent selection process resulted in a smaller but well-characterized cohort. While we acknowledge that a larger sample size would be desirable, we prioritized diagnostic accuracy over quantity to minimize misclassification bias. We definitely agree that expanding the cohort to include additional centers with MRI capabilities would strengthen future studies ( as addressed in the discussion and limitations sections in the revised manuscript). 

Comment 2: Limited applicability to males and atypical TTC variants. Since the scoring system includes "female sex" as a scoring item, there is an inherent gender bias. Its diagnostic accuracy in male patients or atypical TTC variants (e.g., midventricular or basal ballooning types) remains unclear and unvalidated.

 Regarding the limitations of our scoring system in male patients and atypical TTC variants. In our study the inclusion of female sex as a scoring item was similar to previously described scores in literature as mentioned in the manuscript. However, taking in concern that TTC is epidmiologically more prevalent but no exclusive in females. We recognize that this introduces inherent gender bias in the scoring system. We agree that further validation in larger male cohorts is required for more generalization of the score. Regarding the atypical variants, our score didn’t address the various echocardiographic features of TTC, the main purpose of our study was to differentiate TTC from ACS in the acute setting (as described in the limitations section of the revised manuscript) 

Comment 3: Lack of head-to-head comparison with existing scores. The study does not adequately compare the proposed scoring system with other established diagnostic tools such as the InterTAK Diagnostic Score, making it difficult to determine its relative clinical superiority. Could authers calculate InterTAK score in patients of this study?

Concerning calculation of InterTak score in our cohort. The InterTAK score consists of  7 scoring items that adds up to 100 points, it consists female gender (25 points), emotional stress (24 points), physical stress( 13 points), absence of STD (12 points), psychiatric disorder (11 points), neurological disorder (9 points), prolonged QTc (6 points). our study is a retrospective study limited by the available data in medical records, we were not able to recruit information regarding psychiatric or neurological disorders and thus it is not possible to calculate the InterTak score. 

Round 2

Reviewer 1 Report

Comments and Suggestions for Authors

Takotsubo cardiomyopathy (TTC) is still being used in the manuscript instead of TS - see Conclusion. Please check and edit accordingly.

Author Response

Thank you for your remark. We checked and edited accordingly. 

Reviewer 3 Report

Comments and Suggestions for Authors

The authors have accepted all the suggestions and significantly improved the quality of the manuscript.

Author Response

Thank you for your remarks.